# Immobilization of Horseradish Peroxidase on Magnetite-Alginate Beads to Enable Effective Strong Binding and Enzyme Recycling during Anthraquinone Dyes’ Degradation

**DOI:** 10.3390/polym14132614

**Published:** 2022-06-28

**Authors:** Marko Jonović, Branimir Jugović, Milena Žuža, Verica Đorđević, Nikola Milašinović, Branko Bugarski, Zorica Knežević-Jugović

**Affiliations:** 1Institute of Chemistry, Technology and Metallurgy, University of Belgrade, Njegoševa 12, 11000 Belgrade, Serbia; marko.jonovic@ihtm.bg.ac.rs; 2Institute of Technical Science of the Serbian Academy of Sciences and Arts (SASA), Knez Mihailova 35/IV, 11000 Belgrade, Serbia; branimir.jugovic@itn.sanu.ac.rs; 3Department of Biochemical Engineering and Biotechnology, Faculty of Technology and Metallurgy, University of Belgrade, Karnegijeva 4, 11000 Belgrade, Serbia; mzuza@tmf.bg.ac.rs; 4Department of Chemical Engineering, Faculty of Technology and Metallurgy, University of Belgrade, Karnegijeva 4, 11000, Belgrade, Serbia; vmanojlovic@tmf.bg.ac.rs (V.Đ.); branko@tmf.bg.ac.rs (B.B.); 5Department of Forensic Engineering, University of Criminal Investigation and Police Studies, Cara Dušana 196, 11080 Belgrade, Serbia; nikolla.milasinovic@kpu.edu.rs

**Keywords:** horseradish peroxidase, anthraquinone dye decolorization, alginate beads, magnetite nanobeads, covalent immobilization, electrostatic extrusion, wastewater treatment

## Abstract

The aim of this study was to investigate covalent immobilization of horseradish peroxidase (HRP) on magnetic nanoparticles (Mag) encapsulated in calcium alginate beads (MABs) for color degradation, combining easy and fast removal of biocatalyst from the reaction mixture due to its magnetic properties and strong binding due to surface alginate functional groups. MABs obtained by extrusion techniques were analyzed by optical microscopy, FEG-SEM and characterized regarding mechanical properties, magnetization and HRP binding. HRP with initial concentration of 10 mg/g_carrier_ was successfully covalently bonded on MABs (diameter ~1 mm, magnetite/alginate ratio 1:4), with protein loading of 8.9 mg/g_carrier_, immobilization yield 96.9% and activity 32.8 U/g. Immobilized HRP on MABs (HRP-MABs) was then used to catalyze degradation of two anthraquinonic dyes, Acid Blue 225 (AB225) and Acid Violet 109 (AV109), as models for wastewater pollutants. HRP-MABs decolorized 77.3% and 76.1% of AV109 and AB225, respectively after 15 min under optimal conditions (0.097 mM H_2_O_2_, 200 mg of HRP-MABs (8.9 mg/g_carrier_), 0.08 and 0.1 g/mg beads/dye ratio for AV109 and AB225, respectively). Biocatalyst was used for 7 repeated cycles retaining 75% and 51% of initial activity for AB225 and AV109, respectively, showing potential for use in large scale applications for colored wastewater treatment.

## 1. Introduction

Synthetic dyes have a complex aromatic molecular structure and at least one chromophore group that possesses coloration and are a necessity in various significant industries such as leather, paper, as well as textile industries for their color-giving properties [1]. The textile dyes are particularly severe pollutants due to their high toxicity to the environment, as well as the potential toxicity that is a result of dye degradation in the ecosystem [2]. Although the composition of the textile dye degradation products varies, highly mutagenic or carcinogenic compounds can arise during the reduction of dyes and their intermediates. Consumption of such dye-contaminated water by humans can cause a variety of adverse health effects such as breathing problems, wide-ranging immune suppression, central nervous system disorders, behavioral problems, allergic reactions, tissue necrosis, and infections of the skin and eyes [3]. These environmental and health concerns have increased industrial interest in finding effective methods for the removal or degradation of large quantities of these polluting effluents from wastewater in a short time span without producing secondary pollution [4].

Various dye removal methods have been established in countless research papers claiming successful dye removal, such as physical adsorption [5], chemical oxidation [6], chemical coagulation/precipitation [7], electrochemical oxidation [8,9], ultra/nano filtration processes [10], photocatalysis [11], and biological anaerobic/aerobic degradation and/or conversion [12]. So far, there have been a number of shortcomings of synthetic dye removal techniques that must be conquered: large volumes of activated sludge whose proper storage and treatment is required, together with expensive and inefficient processes. Bioremediation of synthetic dye wastewater is the latest approach in addressing this environmental problem. It is classified as a completely natural method where organic dyes are converted into completely harmless end products [13].

Horseradish peroxidase (HRP) is a heme containing enzyme capable of catalyzing the oxidation of various organic compounds such as chlorinated organic compounds, phenols, textile industry dyes, effluents and oil [14] with simultaneous reduction of its co-substrate hydrogen peroxide (H_2_O_2_), and creates large reactive hydroxyl radicals. Especially, HRP as a bioremediation catalyst has shown potential for use because of its ability to oxidize a large number of dyes even in the presence of contaminants commonly found in wastewaters, its ability to function at wide ranges of temperature and pH, its availability and relatively low cost [15,16,17,18]. However, under various conditions HRP is not stable. The main problem in the reaction catalyzed by free HRP is the cost of the enzyme as well as the low stability of the biocatalyst, which significantly increases the cost of the whole process. One approach to increase the stability of the enzyme and develop a cost-effective bioprocess could be to immobilize the biocatalyst on particles and thereby enable its re-use [19,20]. With enzyme immobilization we can endow enzymes with advantageous properties. The immobilized enzymes can be easily separated from reaction systems for reuse and used repeatedly or continuously in a variety of reactors, which would make the work-up simple and the final product uncontaminated [21,22]. Higher selectivity and specificity have been also often reported for immobilized enzymes [23].

Various immobilization carriers such as macroporous exchange resins, nanoporous silica gel, hydrophobic sol-gels, chitosan beads, polyacrylamide, polyvinyl alcohol, calcium-alginate beads and magnetic materials have been used for peroxidase immobilization [24]. The hydrogels and nanomaterials which provide large surface area, a large number of functional groups and other benefits are not optimal carriers in terms of recovery and reuse using common separation unit operations such as centrifugation or filtration. However, magnetic separation utilizing magnetic nanobeads may become a promising method for the separation and recovery of the biocatalyst due to the rapid separation from the medium and easy manipulation with an external magnetic field.

As a magnetic material, iron oxide nanoparticles (Mag) have been successfully used not only for enzyme immobilization but also in many other fields such as separation of biochemical products and magnetically assisted drug delivery [25]. They can be modified with functional groups or inorganic compounds to obtain a magnetic carrier with promise for enzyme immobilization [26]. There are a few studies showing that such an HRP-based magnetic biocatalyst has strong potential in removing organic pollutants [27,28]. In particular, physical immobilization of HRP on Mag improved stability toward the denaturation induced by pH, heat, metal ions, urea, detergent, and water–miscible organic solvent [26]. For example, HRP immobilized on NH_2_-modified magnetic Fe_3_O_4_/SiO_2_ particles showed rapid degradation of 2,4-dichlorophenol [28]. Even more, there is evidence that the presence of magnetite particles increases the activity of horseradish peroxidase [29]. Namely, the rate of reaction of horseradish peroxidase in the enzyme assay increased 30-fold in the presence of magnetite particles (Fe_3_O_4_). It is hypothesized that presence of magnetic particles in the assay has an effect on random-encounter diffusing radicals (F-pairs) via an accelerated intersystem crossing mechanism and the spin states of geminate pairs (G-pairs) which can alter the overall kinetics of the enzyme reaction [29,30].

However, Mag have some limitations in practical application, such as instability (easily oxidized in air), easy agglomeration, problematic distribution and collection [31]. Integrating Mag into other materials to create composite or hybrid materials (e.g., carbon nanotubes with Mag, biopolymers conjugated with Mag) [32,33,34] or coating of Mag with some polymer (e.g., alginate, polyacrylic acid, poly(vinyl alcohol), polyethyleneimine, poly(isobutylene-alt-maleic anhydride) polyethylene glycol, starch, methoxypoly(ethylene glycol)) [35,36,37,38,39] are the main strategies used for solving these shortcomings of Mag. Herein, we propose embedding Mag into hydrogel mm- or μm-scale particles (beads) which combine advantages of both (gel particles and Mag)–rapid separation by external magnetic field and/or gravitational settling due to high density, stability against dissolution especially at lower pH of solution, amplified enzyme activity and a number of available methods able to give large amounts of beads under mild and simple preparation conditions. Of our particular interest is entrapment of Mag in calcium alginate beads, since covalent binding of enzymes to calcium alginate using carbodiimide chemistry has been described as a suitable method to achieve stabilization and good performance of several enzymes including alcalase [40], β-galactosidase [22], acetylcholinesterase [23], α-amylase [24] and others. According to our knowledge, this is the first time that peroxidase is covalently immobilized on Mag-alginate beads. The closest system to ours is the one given by Wasak et al. who used DyP-type peroxidase for immobilization into alginate ferromagnetic beads, but the authors applied physical entrapment as the immobilization procedure [41].

This study results from the necessity to develop an optimal HRP-immobilized catalyst regarding Mag-alginate ratio, bead size and immobilization conditions, which will have the best characteristics in terms of the yield of immobilization, enzyme binding capacities, mechanical stability and activity of the immobilized enzyme. The performance of the optimal catalyst should be tested in the industrially feasible reactions; hence, the efficiency of the biocatalyst in removing two dyes, AB225 and AV109, has been assessed on the basis of decolorization and reusability. These acid anthraquinone dyes, widely used for the dyeing and printing of wool, polyamide, silk and blended fibers, have been selected as models for the current study as a major class of environmental colored pollutants due to their high toxicity, and resistant to degradation making it obligatory to remove them from industrial effluents before the latter are discharged into the environment [42].

## 2. Materials and Methods

### 2.1. Materials

Anthraquinone dyes used in this paper, C.I. Acid Blue 225 (AB225) and C.I. Acid Violet 109 (AV109), were obtained from Lanaset (Lanaset Violet B, Lanaset Blue 2R, Huntsman International LL, The Woodlands, TX, USA). HRP peroxidase (EC 1.11.1.7; donor: H_2_O_2_ oxidoreductase) with a specific activity of 250 purpurogallin units per mg, pyrogallol, 2-mercaptoethanol, and hydrochloric acid were obtained from Sigma-Aldrich (St. Louis, MO, USA). Tris(hydroxymethyl)-methylamine and sodium alginate of low viscosity were obtained from Thermo Fisher Scientific (Waltham, MA, USA). Iron (III) chloride hexahydrate was obtained from Analytika (Prague, Czech Republic). Hydrogen peroxide (H_2_O_2_) 3% (0.97 M) was purchased from Galafarm (Belgrade, Serbia). Calcium chloride and iron (II) sulfate heptahydrate were obtained from Lach-Ner (Neratovice, Czech Republic). Sodium-dihydrogen phosphate, disodium-hydrogen phosphate and acetic acid were obtained from Zorka Pharma (Šabac, Serbia). 1-ethyl-3-(3-dimethylaminopropyl) carbodiimide (EDAC) was obtained from Thermo Fisher Scientific (Waltham, MA, USA). Phenolphthalein indicator was obtained from Molar Chemicals KFT (Budapest, Hungary). Ammonium hydroxide was obtained from Centrohem (Stara Pazova, Serbia). Other chemicals used in this work were of commercial analytical grade.

### 2.2. Synthesis of Magnetic Particles

Magnetic particles were synthesized by the co-precipitation of FeCl_3_ and FeSO_4_ in an excess of NH_4_OH. In each experiment, 5.6 g FeSO_4_ × 7H_2_O and 10.8 g of FeCl_3_ × 6H_2_O were dissolved in 300 mL of double-distilled water, and then heated at 80 °C. A solution of 200 mL of 25% NH_4_OH was added with vigorous and continuous stirring for 60 min. The magnetic particles were filtered and further treated as previously described [43,44]. The obtained particles were thoroughly washed with double-distilled water to pH 7, dried at 105 °C overnight, and then sieved (<100 µm). The obtained particles were treated with 100 mL solution of phenolphthalein at 50 °C for 24 h, and washed with double-distilled water and then dried again overnight.

### 2.3. Preparation of Mag-Alginate Beads (MABs)

MABs were obtained by extrusion droplet generation, with and without electric generator [45]. Polymer solution of 2% (*w*/*v*) was prepared by dissolving low viscosity sodium alginate powder in distilled water. Then the suspension was agitated for 3h by a magnetic stirrer (500 rpm) (Staufen, Germany). Different mass (0.04–0.2 g) of magnetite particles was added to 20 mL of 2% (*w*/*v*) natrium alginate solution. Spherical droplets were formed by extrusion of the polymer suspension through a blunt stainless-steel nozzle (18–25 gauge) using a syringe pump (Razel, Scientific Instruments, Stamford, CT, USA) and a 20 mL plastic syringe. The distance between the nozzle tip and the hardening solution (1.5% (*w*/*v*) CaCl_2_ solution) was 2.5 cm while the flow rate of polymer solution was kept at 27.3 mL h^−1^. For the electrostatic extrusion, electrode geometry with the positively charged nozzle and a grounded hardening solution was applied. The potential difference was controlled at 6.5 kV by a high voltage DC unit (Model 30R, Bertan Associates, Inc., New York, NY, USA). Beads were washed with distilled water and stored at 4 °C in 50 mM Tris-HCl buffer, pH 8.5 before being used. Alginate beads in the absence of magnetite were also formed under the same conditions. The only difference in the method was that magnetite particles were not added to the sodium alginate solution.

### 2.4. Characterization of MABs 

Data related to bead size were obtained using optical microscope Olympus CX41RF, equipped with picture analyzing software ‘‘CellA’’ (Olympus, Tokyo, Japan). All beads were examined immediately after formation. In each case, the sizes of 100 particles were measured.

The surface morphology of MABs was examined using scanning electron microscopy (FEG–SEM) which was performed with field emission gun TESCAN MIRA3 electron microscope at an accelerating voltage of 10 kV. Prior to the observation the samples were degassed and sputter coated with gold using a Polaron SC502 Sputter Coater (Fison Instruments, Glasgow, UK).

The mechanical strength of the beads was measured by compression testing of beads between two flat surfaces. The tests were performed using Universal Testing Machine, AG-Xplus (Shimadzu, Kyoto, Japan) equipped with a 100 N force load cell (force range from 0.01 to 100 N). The compression was performed up to a 30% of sample deformation at compression speeds 0.25 mm/min. Software TRAPEZIUMX (version 1.13, Shimadzu, Kyoto, Japan) was used for recording and analyzing the experimental data.

The magnetic sizes of the magnetic particles and MAB beads were measured using Quantum Design MPMS (Magnetic Property Measurement System) XL-5 SQUID magnetometer which can measure the magnetization of the sample in the interval of temperature 1.8 to 350 K and the range of magnetic fields from −50 to 50 kOe. The magnetic hysteresis loops of Mags and MABs were measured using a vibration sample magnetometer at 5 K. Magnetic characterization of dried beads was performed with a commercial Quantum Design SQUID MPMS magnetometer (Quantum Design, San Diego, CA, USA). The magnetization curve at 5 K of a single bead was measured as a function of the magnetic field between 0 Oe and 50 kOe [46,47].

Fourier transformation infrared spectroscopy (FTIR) analysis was performed using a Nicolet iS 10 spectrometer (Thermo Fisher Scientific, Waltham, MA, USA) in the attenuated total reflectance (ATR) mode at 4 cm^−1^ resolution with ATR correction and OMNIC software (Thermo Fisher Scientific, Waltham, MA, USA). The spectra were obtained in the wave number range between 4000 and 400 cm^−1^ at 25 °C. FTIR was used to verify the chemical interaction and changes after the immobilization.

### 2.5. Immobilization of HRP on MABs

For the immobilization of HRP on MABs the optimal conditions from our previous work have been used [40]. To activate 0.5 g of MABs, they were treated with 10 mg of EDAC in 10 mL of 50 mM Tris-HCl buffer pH 8.5 for 30 min under gentle stirring (150 rpm) at 25 °C. After reaction of the MABs with EDAC, further reactivity of EDAC with carboxyl groups was quenched by addition of 2-mercaptoethanol (10-fold in excess of EDAC) for 10 min. The beads were filtered and excess 2-mercaptoethanol was removed by rinsing with the reaction medium which prevented EDAC from reacting with the active site carboxyl groups of the enzyme during the protein immobilization step [48]. Then 2 mL (12.5–2500 mg/L) of enzyme solution was added to the activated beads with 18 mL of 50 mM Tris-HCl buffer pH 8.5 and immobilization continued at 25 °C for 20 h under gentle stirring (150 rpm). The produced derivative was washed with 50 mM Tris-HCl buffer, pH 8.5, followed by washing with distilled water, after which it was stored at 4 °C in 50 mM Tris-HCl buffer, pH 8.5 before being used. Samples of the enzyme solution before and after the immobilization, together with the washing solutions, were taken for protein content determination. HRP concentration was determined according to the Bradford method [49]. The amount of bound enzyme was determined indirectly from the difference between the amount of enzyme introduced into the coupling reaction mixture and the amount of enzyme in the filtrate and in the washing solutions. The efficiency of immobilization was evaluated in terms of enzyme coupling yield. The enzyme coupling yield was calculated as follows:(1)$$\eta_{\mathrm{enz}}=\frac{P_{g}}{P_{0}}\times 100$$
where *P_g_* is the immobilized amount of protein and *P*_0_ is the initial amount of protein in the enzyme coupling solution determined by the Bradford method. The activity was determined using the standard substrate.

For a comparison purposes, HRP has also been encapsulated into alginate beads with magnetite. In 100 mL of 2% sodium-alginate solution in 50 mM Tris-HCl buffer, pH 8.5, a mass of 50 mg of EDAC has been added and stirred on the magnetic stirrer (150 rpm) for 30 min at 25 °C. Then, 6.25 mg of HRP has been added and stirred (150 rpm) at 25 °C for 20 h. After that period, the addition of 0.5 g of magnetite to the solution has been followed by stirring to uniformly deploy magnetite through the solution (initial enzyme/ carrier ratio was 2.5 mg/g). Spherical droplets were formed by electrostatic extrusion of the polymer suspension through a blunt stainless-steel nozzle (22 gauge) using the syringe pump and a 20 mL plastic syringe. The electrostatic extrusion was performed using the previously described method and equipment. The obtained beads were washed with distilled water and stored at 4 °C in 50 mM Tris-HCl buffer, pH 8.5 before being used.

### 2.6. Activity Assay of Free and Immobilized HRP by Different Methods

The activity of the free and immobilized HRP was measured first using the standard method with substrate pyrogallol and H_2_O_2_ in a reaction medium [50] consisting of 3 mL of 0.013 M pyrogallol solution in the potassium phosphate buffer (pH 7.0; 0.1 M). Briefly, 2 mg of immobilized enzyme (0.05–10 mg HRP/g_carrier_) and 30 µL of H_2_O_2_ (3% *v*/*v*) were placed in the cuvette. By adding 30 µL of H_2_O_2_ (3% *v*/*v*), the reaction was started. The oxidation of pyrogallol (yellow) by HRP-MAB to purpurogallin (dark brown) was monitored at 420 nm absorbance (ε(420 nm) = 12 mM^−1^ cm^−1^) using a UV–Vis spectrophotometer (UV Shimadzu 1700, Shimadzu Corporation, Kyoto, Japan). The change in absorbance in the sample, against the reagent blank, was recorded each 30 s for 3 min. One unit of activity was defined as the amount of biocatalyst that formed 1.0 mg of purpurogallin from pyrogallol in 20 s at pH 7.0 and 20 °C [51]. The activity of HRP-MABs immobilized under optimal conditions was then compared to the activity of the HRP covalently immobilized on the alginate beads in the absence of magnetite under the same conditions, encapsulating HRP into Mag-alginate beads with the same initial enzyme/carrier ratio and free HRP using the standard method.

### 2.7. Optimum pH, Optimum Temperature and Thermal Stability

The optimum pH and temperature for free and HRP-MABs were assessed using standard protocol and pyrogallol as substrate, as previously described. The following buffers have been used for pH optimum assay: 200 mM acetic buffer for pH 3–6 and 100 mM Tris-HCl buffer for pH 7–9. The optimum temperature for free HRP and HRP-MABs was determined using pyrogallol as substrate and varying the temperature in the range 25–55 °C in 100 mM potassium phosphate buffer, pH 7.0. In thermal stability assay, free or HRP-MABs were incubated in the buffer (pH 7) at 60 and 70 °C for 30–120 min before addition of H_2_O_2_ and pyrogallol. After cooling, the enzyme activity was instantly assessed under standard assay conditions. The highest value of enzyme activity of both free and HRP-MABs in each set of experiments was taken as a reference 100% activity.

### 2.8. Dye Removal Efficiency and Stability of HRP-MABs

The removal efficiency of two anthraquinone dyes (AB225 and AV109) from an aqueous solution by the immobilized enzyme was investigated. In order to optimize process parameters for dye decolorization from the simulated textile wastewater with HRP-MAB, the influence of enzyme concentration, dye concentration, H_2_O_2_ concentration, mass of the HRP-MABs, MAG-alginate ratio and particle size of the HRP-MABs was examined for both dyes. The dye (AV109 or AB225) was prepared in 0.2 M acetic buffer, pH 3.6 at concentrations of 10–100 mg/L. The solution was allowed to achieve thermal equilibrium, prior to reaction initiation and was stirred at 100 rpm using magnetic stirring bars. A volume of 3 mL of dye solution was transferred to the cuvette and 50–500 mg of HRP-MABs was added to the solution. The reaction was initiated by the addition of an appropriate amount of H_2_O_2_ to obtain final concentration in the range 0.097–0.97 mM. Samples were analyzed every 5 min using UV–Vis spectrophotometer (UV Shimadzu 1700, Shimadzu Corporation, Kyoto, Japan) at the maximum wavelength for the tested dye (λ_max_ = 590 nm for AV109, and λ_max_ = 628 nm for AB225) [52]. The preliminary experiments showed that the HRP-MABs alone without H_2_O_2_ decolorized 5–11% of both dyes. Moreover, solutions of both dyes appeared to be stable upon exposure to H_2_O_2_ alone, revealing that the decolorization was a result of both H_2_O_2_ dependent and enzyme dependent reaction.

Percent of decolorization was calculated as follows [53]:(2)$$Decolorization\;\%=\left[\frac{\left(A_{0}-A_{t}\right)}{A_{0}}\right]\times 100$$
where *A*_0_ is the initial absorbance of untreated dye solutions (control) and *A_t_* is the absorbance of dye solutions after enzymatic treatment.

### 2.9. Reusability of the Immobilized Enzyme

At the end of each decolorization reaction, HRP-MABs were separated from the reaction medium by magnets, washed with the immobilization buffer twice to remove any remaining substrate or a product and used in the next catalytic cycle in fresh medium. The reusability study was repeated for seven additional cycles.

### 2.10. Statistical Analysis 

All experiments in this research were carried out in triplicate and results were expressed as means with standard deviation. All the tests were considered statistically significantly at *p* < 0.05. All statistical analyses including calculations were conducted using MathWorks MATLAB software (MATLAB R2021a., Natick, MA, USA). The graphs were created in OriginPro 2019b, (OriginLab Corporation, Northampton, MA, USA).

## 3. Results and Discussion

In this research, magnetite nanoparticles were encapsulated in alginate beads resulting in a hydrophilic coating with carboxyl groups, which could be suitable for HRP covalent binding. Easy separation of the alginate beads from the reaction medium using a conventional magnet can be achieved by introducing ferromagnetic particles into the structure of the polymer [54,55]. The immobilization of HRP by covalent bonding, with EDAC as the activation agent, on MABs was carried out. The schematic illustration of the magnetite-alginate-HRP conjugation through amide bond between the amino group of enzyme and carboxyl group of alginate mediated by EDAC is presented (Figure 1).

### 3.1. Characterization of MABs and HRP-MABs

#### 3.1.1. Beads Characterization

In this study, alginate beads were produced by dispersing a sodium Mag-alginate suspension into droplets, which were afterwards solidified by chemical crosslinking with calcium ions. By variation of the nozzle diameter (18–25 G), beads with a wide diameter range (0.38–3.3 mm) were produced, and as a rule, a reduction in nozzle diameter led to smaller beads. However, for production of beads below 2 mm, an electric field had to be applied during the dropping process and this technique has been generally known as electrostatic extrusion. The mean diameters of the beads obtained by simple dropping and electrostatic enhanced dropping, prior to and after immobilization with HRP are given in Table 1 for the sample with weight ratio Mag-alginate 1:4. It seemed that wet beads obtained by simple dropping presented a spherical form and mean diameter was in the range from 3268.0 ± 32.2 µm to 2408.1 ± 40.7 µm, and from 3309.1 ± 60.3 to 2513.1 ± 54.8 µm for MABs and HRP-MABs, respectively, depending of nozzle size. As can be seen, the beads increased in size upon enzyme immobilization relative to their blank counterparts (Table 1), and this effect (increase of 5–28% relative to the diameter before immobilization) was most pronounced for micro-sized electrostatically generated beads, which correspond to higher immobilization yield (Figure 1). The diameter of the beads had a standard deviation in the range 1–24%, with no systematic dependence on the composition. Heterogeneous size is a known disadvantage of extrusion techniques when a high flow rate of spray regime is applied.

In Figure 1. The images of microbeads (1:4 Mag-alginate ratio, created with 22G_ee_ nozzle) in calcium chloride solution obtained by optical microscope before and after immobilization of HRP are presented. According to images with 5× magnification, both empty MABs and HRP-MAB wet particles (with a diameter of ~500 μm) contained dark dots, which were formed by aggregated Fe_3_O_4_ nanoparticles (with a diameter of ~15 nm), suggesting that the magnetic phase was homogeneously distributed throughout the particle volume. After surface beads activation by EDAC and HRP immobilization steps, the beads showed visible differences of color and shape, turning black from transparent and the immobilized beads appeared to deviate from the ideal sphere. Namely, before immobilization, the MABs were roundly shaped in contrast to immobilized beads, which were slightly deformed from a perfect sphere in a polyhedral shape and which were not transparent, being covered by enzyme on the surface. In Figure 1b, the enzyme-covered beads at first sight appeared to repel each other. However, after closer examination at increased magnification, an outer transparent layer is visible which presumably corresponds to the surface immobilization of the enzyme. The MABs are distorted from perfect spheres probably as a consequence of the treatment by Tris-HCl buffer at pH 8.5 used for HRP immobilization and beads storage for taking pictures under an optical microscope. Namely, in a basic environment, water uptake by calcium alginate occurred which resulted in swelling; generally, the swelling process at basic pH can last for few hours, and then dissolution or degradation begins [56]. Furthermore, the enzyme-covered beads did not touch each other as they were affected by repulsive forces caused by the immobilized enzyme but they preserved their magnetization when a magnet was used (Figure 1b). These differences also showed the successful realizing of the enzyme immobilization; this was proved by SEM and FTIR analysis with more assurance. 

The obtained MABs and HRP-MABs were oven-dried to evaluate the possibility for production of supported bioactive phase in powder form suitable for industrial application. By comparison of the weights of wet and dried beads, the moisture content of the beads was found to be equal to 97.45 ± 0.2%. The SEM images of dry blank MABs and activated alginate beads with immobilized HRP both produced by electrostatic extrusion with 22G are presented in Figure 2. Upon drying, the blank beads shrunk and became partially flattened but kept the round shape (Figure 2a), while the enzyme-covered beads collapsed more and appeared merged (Figure 2b). With 500× magnification views of the respective surfaces, the microscale roughness was observed, especially notable in case of the blank beads while surfaces of the beads with immobilized enzyme appeared as more compact (Figure 2d) compared to enzyme-free beads (Figure 2c). This result can be explained by a lower number of hydrophilic groups on enzyme-immobilized microbeads, since covalent immobilization of HRP occurred via carboxyl groups of MABs. Larosas et. al hypothesized that tannase hydroxyl groups formed intermolecular hydrogen bonds with alginate carboxyl groups and reported a more compact surface of tannase loaded calcium-alginate beads versus empty alginate beads, similar to our observation [57].

#### 3.1.2. Mechanical Properties and Magnetization

The results of compression tests are shown in Figure 3. According to the mechanical strength and elastic modulus, no significant difference was observed between samples of Mag-alginate prepared with different ratios (1:2, 1:3, 1:4, 1:5 and 1:10). The elastic modulus was about 25 KPa. This could be a result of some opposing effects. On the one hand, incorporation of magnetite particles as extra-polymeric components would be expected to disrupt the network and weaken the structure. On the other hand, the addition of magnetite to the alginate solution strengthens the beads’ overall structure, as was proved before that iron-oxide nanoparticles may act as reinforcement entities [47,58]; however, the last remark is true only up to some point. Namely, with increasing in magnetite concentration, the formation of particle agglomerates occurred and they were less rigid than individual nanoparticles, acting as defective points in the polymer matrix, then leading to the decrease in the stiffness [47,58]. Our results are comparable with those of Czichy et al. who reported that Young’s modulus of alginate-methylcellulose gel enriched with magnetite particles did not change significantly with increase in particle concentration up to 35 wt.% [59].

The magnetization curves of Mag and MABs were measured at 5 K of a single bead as a function of the magnetic field between 0 Oe and 50 kOe and are shown in Figure 4. The values of saturation magnetization were found as 80 and 55 emu/g for noncoated Mag and MABs, respectively. It appeared that both curves presented small remanence and coercivity in hysteresis loops, revealing good magnetic properties. The Mag and MAB particles had good water dispersibility and fast magnetic response as they were separated by an external magnet within 1 min (Figure 4b). The decrease in saturation magnetization in the case of the MABs appeared to be due to an additional alginate layer in which the nanoparticles are embedded.

#### 3.1.3. Fourier Transform Infrared Spectroscopy (FTIR) Analysis 

To better characterize interactions between enzyme, alginate and magnetite functional groups, FTIR analysis was conducted and the results are presented in Figure 5. In the high frequency region, the stretching vibrations of O–H bonds of alginate appear in the wavenumber range from 3600 to 3000 cm^−1^ [60], while stretching vibrations of aliphatic C–H are observed at 2950–2850 cm^−1^. The bands around 1632 cm^−1^ can be attributed to the carboxylate ion, the peak at 1460 cm^−1^ to the asymmetric and symmetric stretching vibrations, and those at 1120 and 935 cm^−1^ to the C–O stretching vibration of the ring and the C–O stretching with contributions from C–C–H and C–O–H deformation [56]. The stretching of the C=O group of the undissociated carboxyl group is also detected at 1730 and 1631 cm^−1^, as reported for alginate films [61]. The additional peak at 1414 cm^−1^ is assigned to the symmetric stretching vibration of COO^−^ of sodium alginate [62]. The weak band observed at 1385 cm^−1^ is attributed to the stretching vibration of the C=O bond [63]. The peaks at 1030 cm^−1^, shifted to 1023 cm^−1^ and 1024 cm^−1^ indicated C–O stretching vibrations [64]. Additionally, bands around 800 cm^−1^ can be assigned to mannuronic and guluronic acids, respectively, which are both present in the alginate structure [65,66].

It is noteworthy that the asymmetric stretching vibration of the carboxylate ion (1632–1590 cm^−1^) is shifted to lower wavenumbers in comparison with that expected for sodium alginate [67]. One can see that the absorption region of O–H stretching vibrations of calcium alginate beads is narrower than that of sodium alginate. Such a difference was likely due to the interaction of alginate hydroxyl and carboxyl groups with Ca^2+^ when forming the chelating structure, and to the consequent decrease in the number of hydrogen bonds among OH groups [67].

As far as the FTIR spectrum of the HRP loaded beads is concerned, the fingerprint region (1300–650 cm^−1^) appears to be much richer in characteristic peaks than that of the bare beads. This region includes the contributions from complex interacting vibrations, giving rise to the generally unique fingerprint of each compound. Additionally, the peaks’ shift in wavenumber and decrease in intensity clearly demonstrate that there are interactions among alginate, HRP and Ca^2+^. The FTIR spectra of MAB and HRP-MABs show a peak at 520 cm^−1^ due to Fe–O stretching frequency, while the peaks at 1590 and 1594 are shifted from 1632 cm^−1^ due to O–H bending and stretching vibrations which are attached to the surface of iron atoms [68]. The –OH stretching of HRP loaded microbeads is shifted to lower frequency (3364 cm^−1^), which is due to the electrostatic attraction between the C=O group of polymer matrix and –NH_2_ group of HRP, which confirms that the HRP has been successfully loaded into microbeads [69]. New peaks at 622, 467 and 419 cm^−1^ are observed due to Fe–O group, confirming that the Mags are successfully loaded in the microbeads [70]. A peak is observed between 3417 to 3200 cm^−1^; the broadness of the peak is due to both –OH and –NH stretching frequencies, which indicates that HRP is coated on the microbeads [68].

### 3.2. Covalent Immobilization of HRP

#### 3.2.1. Effect of Immobilization Conditions on Enzymatic Activity and Immobilization Efficiency

The covalent immobilization of HRP onto MABs was carried out by amide bond formation using carbodiimide as a coupling agent. The chemistry involved in HRP immobilization by covalent binding via enzyme amino groups and carboxyl group of alginate is presented in Figure 1. As shown, carbodiimide (EDAC) reacts with carboxyl groups of alginate on MABs forming O-acylisourea intermediate. This intermediate, namely the activated carboxyl compound, reacts promptly with an amino group of the enzyme to form an amide bond and releases an isourea by product (Figure 1).

The performance of the immobilized enzyme was tested on both model substrate (pyrogallol) and on simulated wastewater containing synthetic dyes. The effects of varying bead size (Figure 6a), Mag-alginate mass ratio (1:10–1:2) (Figure 6b), and initial enzyme/carrier ratio (0.05–10 mg/g) (Figure 6c) on the immobilization efficiency (yield and protein loading) and activity of the HRP-MAB biocatalyst were investigated using the pyrogallol assay first to establish an optimal immobilization protocol.

The peroxidase activity of HRP-MAB seemed to vary from negligible to about 32 U/g_carrier_, depending on immobilization conditions. Immobilization yield was in the range 65–98%; as expected, the decrease in bead size led to higher immobilization yield of HRP which was consistent with the increase in activity (Figure 6a). Accordingly, the highest protein loading amounting 9.8 mg/g_carrier_ and the highest activity reaching 32 U/g were achieved with the smallest beads. Other researchers have also reported the increase in activity of immobilized enzyme with decreasing bead size due to lower mass transfer resistance [45,54]. Here it should be stressed that the extrusion process through the narrowest nozzle was accompanied by frequent nozzle clogging which prolonged the MAB production process drastically. Therefore, the beads with an average diameter of about 1 mm (reached after the immobilization process was finished), which expressed the second-best result of immobilization efficiency, were selected as optimal and used in further analysis. As regards the effect of magnetite particles, their increasing amount contributed to enhanced immobilization, higher yield and activity (Figure 6b), but only up to Mag-alginate ratio 1:4. Further increase in the amount of magnetite portion had the opposite effect, perhaps because the carboxyl acid groups of alginate became less exposed for activation due to the presence of magnetite particles in a greater amount on the support surface, thus restricting active sites for enzyme coupling.

Since the covalent attachment of enzyme to a carrier depends not only on bead properties but also on the enzyme concentration, further experiments have been carried out to determine the effects of initial HRP concentration in the coupling solution on protein loading, activity of the immobilized enzyme and immobilization yield. For this purpose, the HRP-MAB biocatalyst obtained under the optimal condition (22 Gee and Mag-alginate 1:4) was used and initial HRP concentration was varied in the range from 12.5 to 2500 mg/L (0.05–10 mg/g enzyme/carrier ratio) (Figure 6c). With an increase in offered enzyme mass, immobilization yield firstly increased rapidly to 96.9%, and then gradual stagnation was detected at a concentration of 2.5 mg/g (enzyme/carrier ratio), indicating the carrier saturation (Figure 6c). Under the best conditions, 8.9 mg of HRP was immobilized on each gram of carrier, achieving a coupling yield of 94.9% and enzyme activity of 32.8 U/g (Figure 6c). These results are similar to or favorable compared with those reported in the literature for HRP immobilized on other carriers. For the purpose of comparison, a maximum HRP binding of 4.8 mg/g and 3.3 mg/g has been achieved on a commercial resin (Purolite® A109) by adsorption and covalent immobilization, respectively [13]. The maximum capacity of HRP adsorption on kaolin has been found to be 4.63 mg/g [53]. However, when comparing the activity of HRP-MAB with values achieved in the literature for the same type of HRP, similar or better results have been described by several researchers, ranging from ~5 to ~85 U/g [13,53]. Steric problems frequently associated with high loadings on the carrier are a possible explanation for discrepancy from some expected values.

The activity of HRP-MABs was compared to the activity of the HRP covalently immobilized on the empty alginate beads and encapsulated HRP in Mag-alginate beads under the same conditions as well as a corresponding amount of free HRP (Figure 7). Encapsulation of HRP into Mag-alginate beads was performed separately under the same conditions (bead diameter 1.03 mm, Mag-alginate ratio 1:4, initial enzyme/carrier ratio of 2.5 mg/g enzyme/carrier). These two types of immobilization were carried out in order to compare the catalytic activities of biocatalysts and determine which biosystem is more efficient in the decolorization process. As expected, because of the exposure of the enzyme to chemical treatment during immobilization and mass transfer limitations in the heterogeneous immobilized system, free HRP had the highest activity of 34.7 U (amount of 8.9 mg), corresponding to specific activity of around 3.9 U/mg). Structural and conformational changes in the enzyme molecule after immobilization could also cause a change in enzyme catalytic properties. As no chemical spacer/agent was involved in the immobilization step, the enzyme molecules might be held too closely to the carrier causing structural changes or decrease the conformational flexibility of enzyme molecules due to the multipoint attachment or decrease the accessibility of the active sites. Furthermore, covalent immobilization can cause an alteration of enzyme microenvironment and partition effects, particularly when the carrier is negatively charged as are alginate beads at the reaction pH value. When retained activity of the immobilized enzyme was analyzed (ratio of specific activity of the immobilized enzyme and of the free one), it seemed that HRP-MABs retained 38.3% of the initial activity.

The encapsulation of HRP into Mag-alginate gave a higher immobilization yield but almost 4 times lower activity than covalently immobilized HRP. The lower activity of the encapsulated HRP might be explained by enzyme overloading and blocking of the active sites of immobilized enzyme at the higher concentrations of enzyme. Furthermore, significant diffusion limitations are common to entrapment immobilization and despite high immobilization yields, reduced enzyme activity and relatively low conversion rates have been reported [41].

Protein loading, immobilization yield and activity for HRP immobilized on empty alginate beads were 8.76 mg/g, 91.2% and 13.1 U/g, respectively, which were slightly lower than for HRP-MABs, at 8.89 mg/g; 94.8%; and 13.3 U/g, respectively. These slightly better values could have been achieved because of the synergetic contribution of the magnetite with the immobilized HRP. Thus, despite the evidence that HRP activity was higher in the presence of magnetite particles [29], there was no significant improvement in the HRP activity in our system. However, magnetic nanoparticles may improve the biocatalyst stability and reusability due to the rapid biocatalyst separation from the medium and easy manipulation with an external magnetic field.

#### 3.2.2. Characterization of the HRP-MABs

##### Effect of Temperature and pH on Free HRP and HRP-MABs

The effect of temperature and pH variation on the activity of free HRP and HRP-MABs was studied and the results are presented in Figure 8. It appeared that the immobilized HRP-MABs presented different pH and temperature profiles relative to the free enzyme. The optimal temperature of HRP-MABs seemed to drop from 45 °C to 35 °C after immobilization (Figure 8a), but the optimal pH moved to the basic side, revealing excellent activity of HRP-MABs under more extreme pH conditions (Figure 8b). The increase of HRP-MABs activity at higher pH could be due to electrostatic interactions between H^+^ from the solution and negative surface charge of the anionic polymer matrix of alginate. The partition effects of H^+^ and OH^−^ could cause a significant difference between the pH of the HRP-MABs microenvironment and buffer solution, resulting in a shift of the optimal pH towards the basic medium. The higher activity of the immobilized HRP at higher pH can also explain different specific activity of HRP-MABs in contrast to the free enzyme since standard assay was performed at pH 7.

Thermal stability experiments were performed with free and immobilized HRP which were incubated in the absence of substrate at 60 and 70 °C and the results are presented in Figure 8c. It seemed that the thermal stability of HRP was significantly improved upon covalent immobilization due to the fact that the covalent bonds between HRP and carrier stabilized the structure of the enzyme molecule even at higher temperature, maintaining its active conformation. Evidently, the HRP-MABs treated at 70 °C for 2 h still held activity of 74.05 ± 0.6%, respectively, whereas the free HRP lost almost 45% of original activity under this condition. This hypothesis is also confirmed by other authors [19,20].

### 3.3. AV109 and AB225 Decolorization

Reaction conditions for the optimization of the AB225 and AV109 degradation process and effects on the decolorization process through time (HRP-MAB size, MAG-alginate ratio, initial HRP concentration, HRP-MABs mass, initial H_2_O_2_ concentration and initial dye concentration) are shown in Appendix A.

#### 3.3.1. Optimization of Process Parameters for AV109 and AB225 Decolorization

The second part of the investigation was devoted to study of the activity and stability of the immobilized HRP in the decolorization reactions of two synthetic dyes, AB225 and AV109 anthraquinone dyes, which belong to a group of barely biodegradable wastes. Generally, it is expected that dye removal using enzyme immobilized on alginate beads may be accomplished by either of two mechanisms, enzymatic biodegradation or bioaccumulation/biosorption of the dye onto alginate beads [54]. Therefore, blank MABs were applied in the decolorization reaction under the same reaction conditions as the immobilized enzyme. A confirmation that the dye degradation with HRP was the predominant mechanism is the fact that MABs were able to remove only 5–11% of dyes. In addition, the solutions of both dyes appeared to be stable upon exposure to H_2_O_2_ alone, revealing that the decolorization was a result of H_2_O_2_ dependent enzymatic reaction.

Free peroxidases show higher decolorization rates for various dyes in comparison to immobilized, but reusability and stability make immobilized peroxidase a more efficient biocatalyst [13,41,52,54]. The literature data regarding the efficiency of immobilized HRP in the treatment of colored wastewater are diverse, ranging from ~70 to ~90 % depending on the immobilized preparation and dye type [13,54].

In this study, all immobilized HRP preparations were tested in decolorization reactions of both dyes and the obtained reaction curves confirmed our conclusions about size- and loading-dependent catalytic behavior of HRP-MABs (Appendix A). The small HRP-MABs particles (1 mm, Mag-alginate 1:4, loaded by using initial enzyme concentration of 10 mg/g enzyme/carrier ratio) enabled the highest conversion rate of 76.3% for AB225 and 75.7% for AV109 under specific reaction conditions (initial dye concentration of 0.1 g/mg beads/dye ratio, H_2_O_2_ concentration of 0.097 mM).

As the next task, the effect of specific reaction conditions (biocatalyst mass, initial dye concentration and initial H_2_O_2_ concentration) on conversion yield was investigated in order to establish the optimal conditions for decolorization reactions with HRP-MABs.

The percentage of dye removal was evidently dependent on the mass of biocatalyst (expressed as g of beads per mg of dye) used in the reaction (Figure 9a). The increase from 0.025 to 0.1 g/mg resulted in more efficient decolorization in the case of both dyes, and the maximum percent of decolorization was 73 and 61% for AV109 and AB225 respectively, attained with 0.1 g/mg of HRP-MABs (equivalent to 0.25 mg of immobilized HRP per mg of substrate) after only 10 min of contact between the beads and dye. Adding more HRP-MABs (0.25 g/mg) caused a diminishing effect on dye removal, which indicated saturation.

The influence of the initial H_2_O_2_ concentration was examined by varying the concentration in the range 0.097–0.97 mM. By increasing the initial H_2_O_2_ concentration, a decrease in decolorization percentage was observed, and optimal concentration of H_2_O_2_ for decolorization reaction was 0.097 mM for both dyes (Figure 9b). It is also evident that the inhibitory effect of HRP co-substrate was more pronounced in the case of AB225 than AV109, since a drop of 42% vs. 12% from a maximal activity was observed with increase of H_2_O_2_ concentration to 0.97 mM.

For the purpose of comparison, kaolin-supported HRP exhibited an optimal H_2_O_2_ concentration of 0.2 mM [53]. Barbosa et al. indicated that an interaction between magnetite and H_2_O_2_ may be responsible for HRP inactivation [71]. Different approaches have been used in order to avoid the inhibitory effect of H_2_O_2_, including the addition of glucose oxidase. Glucose oxidase produces H_2_O_2_ only in doses necessary for the reaction and the inhibitory effect of higher H_2_O_2_ concentration can be avoided [72].

Fused aromatic rings and resonance effects in cyclic structures of anthraquinone dyes make them harder to decolorize than azo dyes. These dyes are recalcitrant chemicals since carbon–carbon bonds have to be broken, which is more difficult to achieve than the breakage of nitrogen bonds in azo dyes due to their lesser electronegativity [73]. Due to the substrate specificity, the effect of initial dye concentration (0.02–0.2 g/mg beads/dye ratio) on decolorization efficiency was examined. From Figure 9c we concluded that tested anthraquinone dyes demonstrated an inhibitory effect on immobilized HRP. Decolorization of both dyes with HRP-MABs increased by increasing dye concentration up to 0.08 and 0.1 g/mg for AV109 and AB225, respectively, then it gradually decreased. Evidently, HRP-MABs showed higher affinity toward AV109 than AB225, and the maximum AV109 concentration that can be decolorized in high percentage (>65%) under a specific condition (1 mm beads, Mag-alginate ratio 1:4, loaded by using initial enzyme concentration of 2.5 mg/g (enzyme/support ratio), H_2_O_2_ concentration 0.097 mM) was 0.04 g/mg (beads/dye ratio).

#### 3.3.2. Reusability of the HRP-MABs and Storage Stability

The reusability of the biocatalyst is one of the most important aspects to consider when designing the immobilized enzyme for large scale applications. We have investigated whether the HRP-MABs could be successfully recycled in repeated batch operations for decolorization of AB225 and AV109 under previously determined optimal reaction parameters.

In order to test the reusability of HRP-MABs, the beads were used for the enzymatic reaction, and at the end of every run, the beads were separated with a magnet, rinsed with deionized water, and transferred to the fresh reaction buffer. The reusability is expressed as relative to the HRP-MABs activity at the first cycle (100%). It was investigated up to seven cycles and decolorization results are depicted in Figure 10. With the increase of repeated operation time from 15 min for the first cycle to 210 min for the last, the catalytic activity of the HRP-MABs decreased slowly. Initially, decolorization results were better for AV109 than AB225, 77.3% and 76.1% (Figure 10a), respectively, but after every cycle the decolorization ratio decreased more rapidly with AV109 than with AB225. After seven cycles, the decolorization rate declined to 75% and 51% of the initial value for AB225 and AV109, respectively (Figure 10b). The substrate or product might cause blocking of some pores of beads that limits the access of dyes to the active site of immobilized HRP after frequent decolorization cycles [54]. Covalently immobilized enzymes usually have long-term operational stability unlike those which are supported by entrapment or adsorption. Intense enzyme leakage due to weak enzyme-carrier interactions and/or porous carrier structure is one of the limitations associated with entrapment and adsorption. The results obtained in this study are somewhat satisfying when analyzing the functional stability of comparable systems. For example, alginate-magnetite beads with entrapped HRP retained ~50% of its original activity after 7 consecutive runs in decolorization of Reactive Blue 5 [41]. HRP immobilized on kaolin maintained 35% of the initial activity after seven successive cycles of AV109 decolorization [53].

The immobilized enzyme was stored at 4 °C during six months and it was observed that the activity in dye degradation did not decrease upon storage. Namely, the HRP-MABs retained almost 100% of the initial activity before storage. Overall, these results demonstrate that although our hypothesis that magnetite nanobeads will improve enzyme mechanical properties has not been confirmed, magnetite has been shown to facilitate separation of the biocatalytic system, enhancing its reusability over repeated use and also improving enzyme storage stability.

## 4. Conclusions

In this study, HRP immobilized on MABs proved to be a promising biocatalyst for degradation of two anthraquinone dyes. HRP-MABs removed 77.3% and 76.1% of AV109 and AB225 under optimal conditions after 15 min, respectively. It has also shown also good reusability after seven cycles, retaining 75% and 51% of the initial degradation activity for AB225 and AV109, respectively. A major limitation in potential commercial application of the immobilized HRP for anthraquinonic dye removal can be the inactivation of the enzyme in the presence of the dyes. Thus, additional investigations and development could be carried out to improve the decolorization process. Despite the limitations, these experimental results indicated that HRP-MABs have a good future in biocatalysis because of low cost, easy production, and efficient removal from the reaction medium for reuse.

## Data Availability

Not applicable.

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
