# Peer review of "Immobilization of Horseradish Peroxidase on Magnetite-Alginate Beads to Enable Effective Strong Binding and Enzyme Recycling during Anthraquinone Dyes’ Degradation"

_polymers, 2022, doi:10.3390/polym14132614_

Round 1

Reviewer 1 Report

Dear Authors

In your manuscript you have investigated covalent immobilization of horseradish peroxi-dase (HRP) on magnetic nanoparticles (Mag) encapsulated in calcium alginate beads (MABs) for color degradation, combining easy and fast removal of biocatalyst from the reaction mixture due to its magnetic properties and strong binding due to surface alginate functional groups. MABs ob-tained by extrusion techniques were analyzed by optical microscopy, FEG-SEM and characterized regarding mechanical properties, magnetization and HRP binding. HRP with initial concentration of 10 mg/gcarrier was successfully covalently bonded on MABs (diameter ~1 mm, magnetite/alginate ratio 1:4), with protein loading of 8.9 mg/gcarrier, immobilization yield 96.9% and activity 32.8 U/g. Then immobilized HRP on MABs (HRP-MABs) was used to catalyze degradation of two anthraqui-nonic dyes, Acid Blue 225 (AB225) and Acid Violet 109 (AV109), as models for wastewater pollu-tants. HRP-MABs decolorized 77.3 and 76.1% of AV109 and AB225, respectively after 15 min under optimal conditions (0.097 mM H2O2, 200 mg of HRP-MABs (8.9 mg/gcarrier), 0.08 and 0.1 g/mg beads/dye ratio for AV109 and AB225, respecitevely). Biocatalyst was used for 7 repeated cycles retaining 75 and 51% of initial activity for AB225 and AV109, respectively, showing potential for use in large scale applications for colored wastewater treatment.

The presented results are interesting for the readers. However, I have main concerns about the achievment of the manuscript goals.

My main concerns can be summarized in the following points:

1- Why the authors did not investigate the HRP immobilized enzyme onto activated alginate beads in the absence of magnetite? This is essential to show the synergetic contribution of the magentite with the immobilized HRP in the declorization process.

2- Why the authors only immobilized the HRP onto the surface of the Mag-Alginate formulated beads? Immobilization of the HRP onto EDCs activated alginate solution first, then mixed with Magentite and finally formulated as biocatalytic beads will give the same final biocomposite BUT with higher immobilization yield and more homogenouesly enzyme distribution. 

3- The authors did not show the effect of the immobilization process and parameters on the retained acctivity of the immobilized enzyme. This point is very critical to show and prove the success of the immobilization protocol.

In conclusion, the authors need to respond on the abovementioned comments  clarify and prove the role of the magnetite before reconsidering their work for publication.

Author Response

My main concerns can be summarized in the following points:

1- Why the authors did not investigate the HRP immobilized enzyme onto activated alginate beads in the absence of magnetite? This is essential to show the synergetic contribution of the magnetite with the immobilized HRP in the decolorization process.

The authors agree that the comparison of the results obtained with horseradish peroxidase immobilized on alginate-magnetite beads (HRP-MABs) with the enzyme immobilized onto activated alginate beads in the absence of magnetite is extremely important and could significantly improve the Manuscript. Thus, we have performed substantial revision of the Manuscript and additional experimental works concerning enzyme encapsulation into magnetite-alginate beads, as well as covalent immobilization of HRP on both magnetite-alginate and empty alginate beads. These experiments have been performed separately and the results have been presented in the revised version (pages 16 and 17). These three types of immobilization protocols were carried out in order to compare catalytic activities of immobilized enzymes obtained by three presented methods and to determine which biosystem is more efficient in decolorization process. Thus, we have added additional paragraph in “2.5. Immobilization of HRP on MABs (page 5), and Figure 7 in the revised version of the Manuscript. The necessary discussion concerning this issue has also been added in Results and Discussion section (pages 16 and 17) and is marked with Track Changes in the revised Manuscript.

2- Why the authors only immobilized the HRP onto the surface of the Mag-Alginate formulated beads? Immobilization of the HRP onto EDCs activated alginate solution first, then mixed with Magnetite and finally formulated as biocatalytic beads will give the same final biocomposite BUT with higher immobilization yield and more homogenously enzyme distribution. 

The immobilization method proposed by our Manuscript is principally covalent immobilization of enzyme on surface of particles, while the procedure proposed by the Reviewer is based on enzyme entrapped in alginate matrix together with magnetite nanoparticles and covalently bonded. Indeed, very high immobilization yield is possible to achieve by entrapment. However, there are also some limitations of the entrapment immobilization. One is enzyme leakage, especially in real application since wastewater may contain high concentrations of ions deteriorating to calcium alginate (sodium, phosphate…) which cause swelling effect and enzyme leakage. Due to covalent bonds between the enzyme and matrix, enzyme leakage (into solution throughout the reaction and during washing of beads after each cycle) presumably would be reduced compared to a system based on immobilization solely by entrapment. However, significant diffusion limitations are common to entrapment immobilization and despite high immobilization yields, reduced enzyme activity and relatively low conversion rates have been reported.  These were our concerns especially in light of the fact that the beads besides enzyme should contain magnetite particles.

Indeed, it is quite original and interesting to investigate a system where the enzyme is covalently bonded within the matrix and compare the obtained results with those presented in our Manuscript. Thus, we have performed an additional experiment and immobilized HRP in Mag-alginate beads at the same conditions (bead diameter 1.03 mm, Mag-alginate ratio 1:4, initial enzyme concentration of 2.5 mg/g enzyme/carrier ratio) and protein loading, immobilization yield and enzyme activity were determined using the standard method with pyrogallol. The obtained results have been added in the Manuscript (Figure 7, page 16 and 17). Further optimization of entrapment protocol could be performed but this would require the whole new set of experiments and extends the limit of one manuscript.

3- The authors did not show the effect of the immobilization process and parameters on the retained activity of the immobilized enzyme. This point is very critical to show and prove the success of the immobilization protocol.

We strongly agree with the Reviewer that the retained activity of the immobilized enzyme is very important to show and prove the success of the immobilization protocol. The results concerning the effect of the immobilization process on the retained activity of the immobilized enzyme have been added as well as biochemical characterization of the immobilized enzyme, HRP-MABs, which is of a high importance for design and application of the immobilized HRP system. The retained activity of the immobilized enzyme has been determined as a ratio of specific activity of the immobilized enzyme and of the free one and this is now clearly written in the revised version and discussed (page 16). As Reviewer 2 also suggested, optimum pH, optimum temperature and thermal stability study has also been performed and the results are inserted in the revised Manuscript. Thus, in the Results and Discussion section new Figure 8a-c, as well as subchapter „3.2.2. Characterization of the HRP-MABs“, have been inserted containing the explanation and comparison of the obtained results (page 17 and 18, subchapter 3.2.2 in the revised version). The Manuscript is consequently significantly improved as the necessary data concerning important biocatalyst performance has been incorporated.

In conclusion, the authors need to respond on the abovementioned comments  clarify and prove the role of the magnetite before reconsidering their work for publication.

The authors would like to thank for useful remarks made by the Reviewer 1 that have improved the Manuscript and hope that each important reviewer’s point was taken into consideration. The authors also hope that the revised version of the Manuscript can be useful for design of an efficient enzymatic process based on immobilized HRP for degradation of anthraquinone dyes’ and its application in wastewater treatment.

Reviewer 2 Report

Dear authors, an article entitled “Immobilization of horseradish peroxidase on magnetite-alginate beads to enable effective strong binding and enzyme recycling during anthraquinone dyes degradation" describe a study on preparation of new magnetic carrier for enzyme immobilization. The manuscript is interesting and well written. The suggestions that could be considered by authors for manuscript quality improvement are listed below.

Comments;

  1. Authors wrote “Interestingly, the enzyme-covered beads did not touch each other as they were affected by repulsive forces caused by immobilized enzyme” This is a very interesting and deeper explanation of this phenomenon should be discussed.
  2. Authors performed a lot of analysis that detailed describe the properties of developed carrier. However, the characteristic of immobilized enzyme compared to soluble form was thoroughly omitted (pH, thermal optima, Kcat etc.). Immobilization often strongly affects the catalytic properties of enzymes and finally influences their further application.
  3. Figure 4 is the additional graph inside the figure that should be more sharp. The visualization of carrier attraction, if is necessary could be supplemented with pictures before the magnet application.
  4. Authors wrote “The –OH stretching of HRP loaded microbeads is shifted to lower frequency (3364 cm-1), which is due to the electrostatic attraction between the C=O group of polymer matrix and –NH2 group of HRP …. This conclusion should be supplemented with appropriate references.
  5. Figure 5; How were normalized ATR-FTIR spectra before their comparison?
  6. Figure 6; The y axis description should be added to all plots.

Author Response

Response to Reviewer 2

Dear authors, an article entitled “Immobilization of horseradish peroxidase on magnetite-alginate beads to enable effective strong binding and enzyme recycling during anthraquinone dyes degradation" describe a study on preparation of new magnetic carrier for enzyme immobilization. The manuscript is interesting and well written. The suggestions that could be considered by authors for manuscript quality improvement are listed below.

 Comments;

  1. Authors wrote “Interestingly, the enzyme-covered beads did not touch each other as they were affected by repulsive forces caused by immobilized enzyme” This is a very interesting and deeper explanation of this phenomenon should be discussed.

In Figure 1b the enzyme-covered beads at first sight appeared as they repel each. However, after closer examination at increased magnification an outer transparent layer is visible which presumably corresponds to the surface adsorption of the enzyme. The MABs are distorted from perfect spheres probably as a consequence of the treatment by Tris-HCl buffer at pH 8.5 used for HRP immobilization and beads storage for taking pictures under optical microscope. Namely, in basic environment water uptake by calcium alginate occurred which resulted with swelling; generally, swelling process at basic pH can last for few hours, and then dissolution or degradation begins (Bajpai and Sharma 2004; Reactive & Functional Polymers 59 (2004) 129–140). Thus, according to the Reviewer comment, the revised Manuscript is improved with this discussion in last paragraph on page 8, and page 9.

2. Authors performed a lot of analysis that detailed describe the properties of developed carrier. However, the characteristic of immobilized enzyme compared to soluble form was thoroughly omitted (pH, thermal optima, Kcat etc.). Immobilization often strongly affects the catalytic properties of enzymes and finally influences their further application.

Authors agree with the comment of the Reviewer that the data concerning biochemical characterization of immobilized enzyme could be very useful for a better comparison to soluble form as well as better prediction and design of the enzymatic process. Thus, we have done additional experiments with HRP-MABs and free enzyme concerning effects of pH and temperature on biocatalyst activity and compared obtained results of optimum pH and temperature with free enzyme. The thermal stability study has also been performed and the results are inserted in Figure 8c. Thus, in the Results and Discussion section new Figure 8a-c, as well as sub-chapter „3.2.2. Characterization of the HRP-MABs”, have been inserted containing the explanation and comparison of the obtained results (page 17, sub-chapter 3.2.2 in the revised version). The necessary methodology regarding this issue has also been added (sub-chapter: 2.7. Optimum pH, Optimum Temperature and Thermal Stability). The Manuscript is consequently significantly improved as the necessary data concerning important biocatalyst performance has been incorporated which is necessary for successful design of the process with the immobilized enzyme.

3. Figure 4 is the additional graph inside the figure that should be sharper. The visualization of carrier attraction, if is necessary could be supplemented with pictures before the magnet application.

Figure 4 has been updated according to the instructions. The Figure is supplemented with pictures before the magnet application (page 13).

4. Authors wrote “The –OH stretching of HRP loaded microbeads is shifted to lower frequency (3364 cm-1), which is due to the electrostatic attraction between the C=O group of polymer matrix and –NH2 group of HRP …. This conclusion should be supplemented with appropriate references.

The relevant reference has been added in the revised version of the Manuscript.

5. Figure 5; How were normalized ATR-FTIR spectra before their comparison?

Thank you for the comment. As it is known the approaches based on original spectra with reference groups and with normalized spectra are based on the idea to remove any variation in the absorbance spectra due to a variation of the IR beam penetration between samples, which would bias further interpretation of results. Although the spectra can be used in their original form, the internal software (OMNIC) of the ATR FTIR instrument does the normalization of the obtained spectra prior to analysis. As the important part, the baseline correction was performed, as well (autocorrected by the internal software). It was all done in the absorbance format because this is in direct proportionality to concentration rather than in transmittance mode. On the other hand, all software for the normalization purposes employ signals that do not change appreciably (most FTIR software usually allow users to upload and overlay the two data files). The OMNIC software itself does the normalization to make the highest value of 1 absorbance unit, enabling us to perform further quantitative analysis. The spectra were then transferred and arranged using Spectragryph free optical spectroscopy software.

6. Figure 6; The y axis description should be added to all plots.

All descriptions for y-axis have been inserted in the Manuscript as well as added to all plots.

The authors would like to thank for useful remarks made by the Reviewer 2 that have improved the Manuscript and hope that each important reviewer’s point was taken into consideration. The authors also hope that the revised version of the Manuscript can be useful for design of an efficient enzymatic process based on immobilized HRP for degradation of anthraquinone dyes’ and its application in wastewater treatment.

Round 2

Reviewer 1 Report

Dear Authors

Thank you very much for considering my comments during the revision of your submitted manuscript.

I can recommend the revised version for publication. 

Author Response

The authors would like to thank Reviewer 1 for the comment and previous valuable remarks regarding our paper, hoping that anthraquinone dyes’ degradation by designed immobilized horseradish peroxidase on magnetite-alginate beads could be considered as an environmentally acceptable process for wastewater treatment.

Reviewer 2 Report

Dear Authors, the manuscript was significantly improved. However some details still need attention.

Comments;

1.       Figure 2; The micrographs could be arranged with box 2x2 and visible particles could be marked by arrows with short  description.

2.       Figure 4 a; The axis are without units.

3.       Figure 5; The spectra plot could be prepared by using Origin pro 2019 with inserting brake in range 2500 to 1950 cm-1 that region do not contains any significant bands. Now the figure looks poorly.

4.       Figure 7; Authors compare activity the free and immobilized HRP. However this comparison is not fully proper. The immobilization can affect many catalytic and operational parameters of enzyme. The comparison should be done not by comparing activity but catalytic efficiency. Authors should determine the Kcat. Moreover the results below shows that free HRP has a different pH optima, compared to immobilized form of enzyme. This can also influence on activity of immobilized enzymes. Authors rightly speculate about mass transfer limitation, but structural changes of enzyme are also main factor that change enzymes catalytic  properties after immobilization. The presented on the plot comparison is a large simplification.

5.       Effect of tepearature and pH on free HRP and HRP-MAB – please correct the typos

Author Response

Responses to the Reviewer 2

Reviewer comment „Dear Authors, the manuscript was significantly improved. However some details still need attention“.

Comments:

  1. Figure 2; The micrographs could be arranged with box 2x2 and visible particles could be marked by arrows with short description.

The micrographs have been arranged, as suggested (page 10 in the revised manuscript).

  1. Figure 4 a; The axis are without units.

The authors thank the Reviewer for pointing out the incomplete information about Figure 4a concerning missing Units scale on both x and y axis. The Figure 4a has been corrected in the revised version of the manuscript, as was suggested by the Reviewer (page 11 in the revised manuscript).

  1. Figure 5; The spectra plot could be prepared by using Origin pro 2019 with inserting brake in range 2500 to 1950 cm-1 that region do not contains any significant bands. Now the figure looks poorly.

The authors thank the Reviewer for the observation. The spectra plot has been improved with inserting brake in range 2500 to 1950 cm-1. Thus, the Figure 5 has been greatly improved since that region did not contain any significant bands (page 13).

  1. Figure 7; Authors compare activity the free and immobilized HRP. However this comparison is not fully proper. The immobilization can affect many catalytic and operational parameters of enzyme. The comparison should be done not by comparing activity but catalytic efficiency. Authors should determine the Kcat. Moreover the results below shows that free HRP has a different pH optima, compared to immobilized form of enzyme. This can also influence on activity of immobilized enzymes. Authors rightly speculate about mass transfer limitation, but structural changes of enzyme are also main factor that change enzymes catalytic properties after immobilization. The presented on the plot comparison is a large simplification.

The authors strongly agree with the Reviewer that the immobilization can affect many catalytic and operational parameters of enzyme and that the limitation in mass transfer is not the only factor influencing the enzyme catalytic properties after immobilization. Structural and conformational changes in the enzyme molecule after immobilization could also cause a change in enzyme catalytic properties, as commented. As no chemical spacer/agent was involved in the immobilization step, the enzyme molecules might be held too closely to the carrier causing structural changes or decrease the conformational flexibility of enzyme molecules due to the multipoint attachment or the change of the microenvironment around the enzyme molecules. Furthermore, the covalent immobilization can cause an alteration of enzyme microenvironment and partition effects, particularly when the carrier is negatively charged as alginate beads at reaction pH value. Thus, this additional discussion which greatly improved the manuscript has been added according to the Reviewer`s comment (page 16: first paragraph, page17: first paragraph).

The authors also strongly agree with the Reviewer that the comparison of activity between free and immobilized enzyme is not fully proper and that could be better by comparing catalytic efficiency not by comparing catalytic activity. In general, the values of kinetic constants like kcat and Km obtained by Michaelis–Menten kinetic model seem to be powerful approach in the characterization of the immobilized enzyme catalytic performances. However, in our previous study, it was found that the kinetics of anthraquinone dyes’ degradation by horseradish peroxidase could not be represented by the simple Michaelis–Menten kinetic model, but by a more complex kinetic model of bisubstrate reactions like the ping-pong bi-bi kinetics with substrate inhibition (Šekuljica, N. et al (2015) Decolorization of anthraquinonic dyes from textile effluent using horseradish peroxidase: Optimization and kinetic study, Scientific World Journal, 2015, art. no. 371625, DOI: 10.1155/2015/371625). Thus, the comparison by kinetic constants including kcat and Km is only valid when a complete kinetic study is done with information about all kinetic constants including inhibition constants and others. Due to this complex character, the kinetic study requires additional experiments, analysis and space in the Journal which can be a little too extensive for one paper. On the other side, the comparison of free and immobilized HRP by retained activity as the ratio of specific activity of the immobilized enzyme and of the free one is a common method in comparison of immobilized enzymes. For example, the following studies only compare the efficiency of both free and immobilized HRP in pollutant removal or only retained activity (M. Petronijevic et al. (2021) Characterization and application of biochar-immobilized crude horseradish peroxidase for removal of phenol from water, Colloids and Surfaces B: Biointerfaces 208: 112038; K. Jankowska et al. (2021) Horseradish peroxidase immobilised onto electrospun fibres and its application in decolourisation of dyes from model sea water, Process Biochemistry 102: 10–21; Šekuljica et al. (2020) Immobilization of horseradish peroxidase onto Purolite® A109 and its anthraquinone dye biodegradation and detoxification potential, Biotechnol Progress 2020;e2991). The next research step could be the kinetic study and comparison of the catalytic efficiency between immobilized and free HRP with respect to the kcat.

  1. Effect of tepearature and pH on free HRP and HRP-MAB – please correct the typos

This has been corrected in the revised Manuscript and marked up using the “Track Changes” function (page 16).

Authors would like to thank the Reviewer 2 for all useful remarks that have significantly improved the manuscript and hope that the revised manuscript could provide contribution in the field of application of an environmentally acceptable enzymatic process for wastewater treatment.
